# Effects of COVID-19 Lockdown on Melanoma Diagnosis in Switzerland: Increased Tumor Thickness in Elderly Females and Shift towards Stage IV Melanoma during Lockdown

**DOI:** 10.3390/cancers14102360

**Published:** 2022-05-10

**Authors:** Lisa Kostner, Sara Elisa Cerminara, Gustavo Santo Pedro Pamplona, Julia-Tatjana Maul, Reinhard Dummer, Egle Ramelyte, Johanna Mangana, Nikolaus Benjamin Wagner, Antonio Cozzio, Saskia Kreiter, Angelika Kogler, Markus Streit, Anja Wysocki, Alfred Zippelius, Heinz Läubli, Alexander Andreas Navarini, Lara Valeska Maul

**Affiliations:** 1Department of Dermatology, University Hospital of Basel, 4031 Basel, Switzerland; lisa.kostner@usb.ch (L.K.); sara.cerminara@usb.ch (S.E.C.); alexander.navarini@usb.ch (A.A.N.); 2Department of Ophthalmology, University of Lausanne, 1004 Lausanne, Switzerland; gsppamplona@gmail.com; 3Department of Health Sciences and Technology, ETH Zurich, 8092 Zurich, Switzerland; 4Department of Dermatology, University Hospital of Zurich, 8091 Zurich, Switzerland; julia-tatjana.maul@usz.ch (J.-T.M.); reinhard.dummer@usz.ch (R.D.); egle.ramelyte@usz.ch (E.R.); johanna.mangana@usz.ch (J.M.); 5Department of Dermatology, Cantonal Hospital of St. Gallen, 9000 St. Gallen, Switzerland; nikolaus.wagner@kssg.ch (N.B.W.); antonio.cozzio@kssg.ch (A.C.); saskia.kre@gmail.com (S.K.); angelika.kogler@stud.medunigraz.at (A.K.); 6Department of Dermatology, Cantonal Hospital of Aarau, 5001 Aarau, Switzerland; markus.streit@ksa.ch; 7Department of Dermatology, Cantonal Hospital of Lucerne, 6000 Lucerne, Switzerland; anja.wysocki@luks.ch; 8Department of Theragnostics, Division of Oncology, University Hospital Basel, 4031 Basel, Switzerland; alfred.zippelius@usb.ch (A.Z.); heinz.laeubli@usb.ch (H.L.); 9Laboratory of Cancer Immunology and Immunotherapy, Department of Biomedicine, University of Basel, University Hospital of Basel, 4031 Basel, Switzerland

**Keywords:** COVID-19, lockdown, melanoma, diagnostic delay, Breslow thickness, age, gender

## Abstract

**Simple Summary:**

Concerns about potential diagnostic delays due to pandemic-related disruptions were expressed early during the COVID-19 outbreak. In the present study, we investigated the impact of the Swiss COVID-19 lockdown between March and June 2020 on newly diagnosed melanomas. Our main findings include a short-term rise in melanoma diagnoses after the major lift of social lockdown restrictions. Further, the analyzed prognostic factor for melanomas, the Breslow thickness, was greater in elderly female patients during the lockdown compared to the pre- and post-lockdown period. In addition, proportionally more stage IV melanomas were diagnosed during the lockdown period. Our study highlights the need for an adjusted management of cancer patients in future pandemic management strategies.

**Abstract:**

At the early stages of the COVID-19 outbreak in 2020, Switzerland was among the countries with the highest number of SARS-CoV2-infections per capita in the world. Lockdowns had a remarkable impact on primary care access and resulted in postponed cancer screenings. The aim of this study was to investigate the effects of the COVID-19 lockdown on the diagnosis of melanomas and stage of melanomas at diagnosis. In this retrospective, exploratory cohort study, 1240 patients with a new diagnosis of melanoma were analyzed at five tertiary care hospitals in German-speaking Switzerland over a period of two years and three months. We compared the pre-lockdown (01/FEB/19–15/MAR/20, n = 655) with the lockdown (16/MAR/20–22/JUN/20, n = 148) and post-lockdown period (23/JUN/20–30/APR/21, n = 437) by evaluating patients’ demographics and prognostic features using Breslow thickness, ulceration, subtype, and stages. We observed a short-term, two-week rise in melanoma diagnoses after the major lift of social lockdown restrictions. The difference of mean Breslow thicknesses was significantly greater in older females during the lockdown compared to the pre-lockdown (1.9 ± 1.3 mm, *p* = 0.03) and post-lockdown period (1.9 ± 1.3 mm, *p* = 0.048). Thickness increase was driven by nodular melanomas (2.9 ± 1.3 mm, *p* = 0.0021; resp. 2.6 ± 1.3 mm, *p* = 0.008). A proportional rise of advanced melanomas was observed during lockdown (*p* = 0.047). The findings provide clinically relevant insights into lockdown-related gender- and age-dependent effects on melanoma diagnosis. Our data highlight a stable course in new melanomas with a lower-than-expected increase in the post-lockdown period. The lockdown period led to a greater thickness in elderly women driven by nodular melanomas and a proportional shift towards stage IV melanoma. We intend to raise awareness for individual cancer care in future pandemic management strategies.

## 1. Introduction

Since the beginning of 2020, the severe acute respiratory syndrome coronavirus 2 (SARS-CoV-2) has caused devastating implications for human health. Many governments announced lockdown strategies to curb the global spread, creating severe impacts on mental health, global economy, educational and healthcare systems [1]. While the health care system focused primarily on SARS-CoV-2 and emergency cases, coronavirus disease 2019 (COVID-19) lockdown measures had a remarkable effect on primary care access. During the pandemic, many countries around the globe postponed or even cancelled their elective cancer screening programs. Recent studies show that lockdown-induced postponement of oncological screenings resulted in an alarming detection drop in multiple cancer entities. Data provided by the Belgian Cancer Registry reported a total decrease of 44% in the diagnosis of invasive cancers in April 2020 compared to the previous year. The largest decline was observed in the diagnosis of non-melanoma skin cancer and melanoma compared to other cancer types, especially in the older population [2]. A significant decrease in melanoma diagnoses during the first lockdown months in 2020 was also observed with concern in the UK and the USA [3,4]. According to a survey by the Global Coalition for Melanoma Patient Advocacy, dermatologists from 36 different countries estimated that one fifth (21%) of melanoma cases remained undiagnosed throughout the COVID-19 pandemic in 2020, implying an estimated 60,000 undiagnosed melanomas worldwide [5]. Moreover, a Greek study found a significantly higher-than-expected percentage of newly diagnosed advanced melanomas in the stages IIC, III and IV after the lockdown in 2020 in comparison to previous years [6]. Similar observations with a rise of more advanced melanomas after lockdown have been reported in several European countries [7,8,9,10].

These pandemic-associated disruptions of cancer care raised worldwide concerns about delays of melanoma diagnosis early on, which may be associated with an increase in morbidity and mortality [11]. An early diagnosis of cancer increases the chances of an effective treatment outcome, especially for melanoma, and skin cancer screenings with dermoscopy are cornerstones in their management [12]. Tumor thickness represents the most important prognostic factor for melanoma [13]. Thus, any delay in diagnosis resulting in increased tumor thickness and ulceration is correlated with the risk of a worsened outcome.

Whether routine skin cancer screenings reduce the melanoma-related burden of disease remains a controversial debate [14,15,16]. The postponement and cancellation of skin cancer screenings during the global pandemic [7,17,18] might, for the first time, provide insight into the value of elective cancer screenings. According to the EADV’s (European Academy of Dermatology and Venereology) melanoma task force for the management of melanoma patients, elective skin cancer screenings for individuals with increased melanoma risk were recommended to be extended by a maximum of 2–3 months during the pandemic [19]. The duration of COVID-19 lockdowns highly varied throughout Europe: 42 days in the first wave in Switzerland, 70 days in Italy, 103 in England, and 112 days in Wales [20].

Even though Switzerland was among the countries with the highest incidence of SARS-CoV-2-infections per capita at the early stages of the COVID-19 outbreak [11], no studies have yet been conducted on the pandemic’s effect on melanoma diagnoses. In the present study, we investigated the impact of the first COVID-19-related lockdown restrictions on melanoma diagnosis in Switzerland by focusing on the divergence in melanoma diagnoses in the pre- and post-lockdown phase. We examined whether there were points in time at which the rate of melanoma diagnoses differed significantly, by inspecting the time series of the moving average of the number of melanoma diagnoses. The aim of the study was also to highlight potential gender- and age-specific differences in prognostic factors of primary invasive melanomas due to the lockdown by analyzing whether Breslow thickness, presence of ulceration, and stage were dependent on the period in which the diagnosis was made.

## 2. Materials and Methods

### 2.1. Study Design and Population

In this retrospective, exploratory cohort study, we obtained clinical data from all adult patients with newly diagnosed invasive primary cutaneous melanoma from five tertiary referral centers in German-speaking Switzerland: University Hospitals of Basel and Zurich, as well as the Cantonal Hospitals of Aarau, Lucerne, and St. Gallen. Exclusion criteria were melanomas in situ and lack of written informed consent which were applied by each center before transferring the according information to the data base. The available data were curated and their quality was assured prior to the analysis.

We analyzed the cases, diagnosed over a period of two years and three months centered on the lockdown onset, using the official date as announced by the Swiss Federal Council when nationwide lockdown measures were established (16/MAR/20) [21]. We collected data from 01/FEB/19 to 30/APR/21 at the considered tertiary referral centers in Switzerland. We used this interval because it comprised data collected during the first lockdown period due to COVID-19 in Switzerland, as well as a considerable amount of data collected before and after the lockdown. We defined three periods of interest in this study: pre-lockdown, lockdown, and post-lockdown. The pre-lockdown period (01/FEB/19–15/MAR/20) was defined as the interval before the lockdown, followed by the lockdown period (16/MAR/20–22/JUN/20) and the post-lockdown period (23/JUN/20–30/APR/21). The lockdown period in Switzerland was defined as a period when severe restrictions were imposed on the gastronomic, educational, commercial, and entertainment sectors, excluding grocery stores and health care facilities. In terms of medical interventions, only emergency and urgent procedures were allowed [22]. The first lift of restrictions in Switzerland (after 42 days) comprised elective medical procedures, such as skin cancer screenings (27/APR/20) [22]. The second lift of restrictions, the largest after the first outbreak in Switzerland, allowed certain social gatherings (e.g., funerals), outpatient treatments and personal services (22/JUN/20) [22]. The data included individual demographic (age and gender), clinical (date of melanoma diagnosis), and histopathological information (melanoma subtype, Breslow thickness, ulceration, stage according to the American Joint Committee on Cancer, 8th edition), provided by the internal clinical and histopathological records.

### 2.2. Data Analysis

#### 2.2.1. Time Series of Diagnosed Melanomas

We investigated whether there were timepoints in which the rate of melanoma diagnoses differed substantially within the whole interval of data collection (01/FEB/19–30/APR/21). The rate of melanoma diagnoses was calculated as the moving 7-day average of diagnosed melanomas within the considered interval. Data points were identified as outliers if they were above or below three standard deviations from the mean [23].

#### 2.2.2. Dependence of Breslow Thickness and Ulceration on the Time of Diagnosis

We used linear mixed models and generalized linear mixed models to investigate the dependence of the Breslow thickness and ulceration, respectively, on the time of diagnosis, as well as the patients’ gender and age. These models were used due to the hierarchical structure of the measurements within each hospital. Hence, we defined Period, Gender, and Age as fixed factors, with Age being a continuous variable, as well as Hospital as the random factor. Period was defined as the pre-lockdown, lockdown, and post-lockdown. We log-transformed the Breslow thickness to not violate the assumption of normality of residuals in linear mixed models. We report interactions containing the factor Period, because of the interest in the dependence of the measures on the lockdown period. Because of the continuous variable Age, we performed post-hoc analyses to determine pairwise differences with Age defined as “young” (mean age minus one standard deviation), “average” (mean age), and “old” (mean age plus one standard deviation), following the convention described in [24,25]. Post-hoc significant differences were identified according to a *p* < 0.05, Tukey-corrected for multiple comparisons. The dependences of Breslow thickness and ulceration on Time were also investigated separately for the melanoma subtypes and stages of diagnosed melanomas (AJCC 8th ed. stage I–IV). Data from subjects with missing relevant information were excluded from each analysis.

#### 2.2.3. Influence of Lockdown Period on the Proportional Distribution of Stage of Diagnosed Melanoma and Corresponding Age

We investigated the dependence of the stage of diagnosed melanoma and the time of diagnosis, i.e., pre-lockdown, lockdown, and post-lockdown. We also investigated the dependence of the stage of melanoma and the patient′s age on the time of diagnosis. Stages of diagnosed melanoma were divided into the stages I, II, III, and IV, according to the AJCC 8th ed. classification. The factor Period was defined as the pre-lockdown, lockdown, and post-lockdown, as described in the analysis in 2.2.2. The factor Age was divided into three subgroups (<58 y.o., 58–72 y.o., and >72 y.o.). The age cut-offs were set according to the terciles of the sample to maintain the age groups with approximately the same number of subjects. Differences in the proportion were considered significant according to chi-square tests and level of significance of 0.05.

## 3. Results

### 3.1. Patient Characteristics

Data from 1240 melanoma patients (741 men [59.8%], mean age = 64.0 ± 15.4 y.o.) were included in the study (Table 1). The Breslow thickness of each patient was collected (median = 1.2 mm [Q1−Q3 = 0.7–2.6], mean = 2.2 mm ± 3.2 mm). Most melanomas were non-ulcerated (914 [73.7%]) and had either a superficial spreading (543 [43.8%]) or a nodular (230 [18.5%]) subtype. 

To illustrate the detected stability of newly diagnosed melanomas during lockdown, we compared equal time intervals to the three-month duration of the lockdown in this sub-analysis. During this 9-month-interval, a total of 445 new melanomas were diagnosed. Within all predefined three equal 3-month-periods, the sample size remained stable with 143 patients out of 445 (32.1%) before lockdown, during the three-month lockdown with 149 out of 445 (33.5%), and 153 out of 445 (34.4%) three months after the lockdown.

### 3.2. Time Series of Melanoma Diagnosis in Switzerland Relative to COVID-19 Lockdown

We observed a significantly higher number of melanoma diagnoses in Switzerland after the lift of major restrictions (Figure 1). In contrast, we observed no significant differences in the rate of melanoma diagnoses right after the lockdown onset (16/MAR/20) or right after the lift of restrictions on medical procedures (27/APR/20). Furthermore, no gender-specific differences were identified in the time series of melanoma diagnoses.

### 3.3. Effects of COVID-19 Lockdown on Thickness and Subtype of Diagnosed Melanomas in Switzerland

We observed a significant interaction between the factors Period, Gender, and Age (F(21,182) = 3.15, *p* = 0.04). Post-hoc analysis showed that the Breslow thickness was significantly greater in older female patients when diagnosed during the lockdown compared to the pre-lockdown (1.9 ± 1.3 mm, *p* = 0.03) and post-lockdown (1.9 ± 1.3 mm, *p* = 0.048) periods in Switzerland (Figure 2A, Table 2). Furthermore, Breslow thickness diagnosed during pre-lockdown was greater in male compared to female patients (1.2 ± 1.1 mm, *p* = 0.04) at average age. Finally, Breslow thickness diagnosed during lockdown was greater in female compared to male patients (1.8 ± 1.3 mm, *p* = 0.019) at old age.

When melanomas were stratified into subtypes, we noticed that the result of greater Breslow thickness in older women was driven by the nodular type of melanoma (Figure 2B, Table 2). We observed a significant interaction between the factors Period, Gender, and Age only for nodular melanoma (F(2,209) = 4.49, *p* = 0.012). Post-hoc analysis showed that the difference in Breslow thickness of newly diagnosed nodular melanomas was greater in older female patients during lockdown compared to the pre-lockdown (2.9 ± 1.3 mm, *p* = 0.0021) and post-lockdown periods (2.6 ± 1.4 mm, *p* = 0.008). At average (*p* = 0.03) and old (*p* = 0.0013) ages, Breslow thickness diagnosed during lockdown was greater in female compared to male patients (1.7 ± 1.3 mm and 3.3 ± 1.4 mm, respectively).

When the melanomas were separated into AJCC 8th ed. stages (Figure 3, Table 1), we observed a significant interaction between the factors Period, Gender, and Age only for stage II melanomas (F(2,173) = 3.39, *p* = 0.04). Post-hoc analysis showed that the Breslow thickness diagnosed during lockdown was greater in female compared to male patients (2.2 ± 1.4 mm, *p* = 0.016) at older age. 

No significant main effects or interactions between factors were detected for the presence of ulceration, considering either all data stratified by melanoma subtype or by AJCC 8th ed. stage (*p* > 0.05).

### 3.4. Stage-Dependent Effects on the Melanoma Diagnosis Relative to COVID-19 Lockdown

The proportional distribution of stages of diagnosed melanoma was different during the lockdown period compared to the other periods considered [χ²(6) = 12.8, *p* = 0.047]. While post-hoc analysis indicated no significant differences, stage IV melanomas diagnosed during the lockdown period exhibited the largest residual (2.30) compared to the other melanoma stages (Figure 4). The age of the patients diagnosed with melanoma was not different across all periods relative to lockdown [χ²(4) = 1.09, *p* = 0.9] (Figure 5). 

## 4. Discussion

### 4.1. Main Findings

In this retrospective, exploratory cohort study, we found several effects of the COVID-19 lockdown period on the diagnosis of melanoma in Switzerland. First, we observed a significantly higher rate of melanoma diagnoses after the lift of major restrictions. Second, we found that the Breslow thickness was diagnosed greater in older female patients during the lockdown compared to no-lockdown periods, and that this result is driven by the diagnosis of nodular melanomas. Finally, we showed that a more advanced stage of melanoma was diagnosed during the lockdown period. 

### 4.2. Perspective

Worldwide concern is growing over an impending cancer pandemic as diagnosis delay is linked to a rising mortality rate during the COVID-19 pandemic [26]. Our study showed an increase in the rate of newly diagnosed melanomas immediately after the lift of major social restrictions in Switzerland. During the rest of our observation period, we observed no significant differences in the rate of new melanoma diagnoses. We assume that due to lockdown measures and the lack of skin cancer screenings, certain patients have experienced a delay, even if not quantified statistically, which has led to the rapid increase in new melanoma cases after the lift of major restrictions in our study. Various studies pointed out that the delay in the diagnosis of melanomas due to COVID-19-related restrictions was associated with a reduction in the number of skin cancer diagnoses of up to 60% [6,27,28]. Among all cancers, cutaneous malignancies were the ones with the highest number of missed diagnoses with a diagnostic decrease of 56.7% in 2020 vs. 2019 [27]. While we observed no differences in the rate of melanoma diagnoses during the lockdown period, dermatologists in London even observed an increased melanoma detection rate. The authors linked the increase in the number of cases to a heightened self-perception of patients [29]. We assume that young patients showed an increased self-perception and would rather consult doctors whereas older patients avoided any social contacts due to fear of infection, leading to the short-term peak of newly diagnosed melanomas immediately after the lift of major restrictions. Whereas our findings demonstrate a quick return to average numbers after the peak, a recent study from the United States described a more constant and slowly growing rebound in consultations, yet without returning to normality [30]. While pandemic restrictions were imposed all over the world, measurements varied from country to country and the impacts of COVID were multifaceted, explaining the regional variety of differences. More severely COVID-19-affected regions such as Northern Italy reported a more significant reduction in melanoma diagnoses [8,31]. A study comparing COVID-19 mortality rates across Europe also highlighted that some countries (e.g., Italy and Spain) had slightly stricter lockdown measurements than others [32]. Similar to our findings, rates of newly diagnosed melanomas did not differ significantly in Southern Italy, justified by less fear in areas with lower COVID incidences [8]. Even though an upsurge in teledermatology was observed during the COVID era, which might have also led to the stable melanoma detection rates in our study, physical consultation remains more accurate in the detection of potentially malignant skin lesions and might countermeasure delays in direct comparison with teleconsultations [33,34]. 

We observed a greater tumor thickness among older female patients when diagnosed during lockdown compared to the pre- and post-lockdown periods. Although from a demographic point of view, gender-related differences in melanoma are well established and suggest a female survival advantage [35,36,37], gender-specific differences in carcinoma diagnosis during medical restrictions due to the pandemic need to be considered with concern. From a socio-demographic perspective, especially older, poorer women under inferior health conditions are more concerned and compliant with rules set by the government [38]. There is evidence that women present a higher level of understanding and agreement regarding the implemented restriction measures compared to men [38], as well as a higher level of fear regarding COVID-19, especially if already actively treated for cancer [39]. However, contrary evaluations revealed an increased female representation among melanoma patients during lockdown [6], suggesting that men seem less likely to engage in preventive behaviors [40] or perform skin self-examination [41]. Higher age was considered an influencing factor, showing a post-lockdown increase in Breslow thickness among older patients (aged > 50 years) [7]. Another study has linked Breslow thickness to patient-related factors such as higher age and living in a nursing home [42]. These findings support our observations of thicker melanomas in elderly people during lockdown, which in Switzerland occurred only in older females. Consequently, based on the observed greater Breslow thickness in older women and previous gender-behavioral studies, we support the hypothesis of a more fearful and reserved attitude in elderly females. Since no gender-related shifts in melanoma prevalence and incidence have been reported before the pandemic in general and especially in Switzerland [43], our study highlights that gender variances under pandemic conditions need further attention. Fear and insecurity in the face of a potentially destructive disease might be one reason for delayed melanoma diagnosis due to its impact on personal prevention and care [44,45]. A recent German questionnaire-based evaluation highlighted the impact of fear by reporting that out of all hospital-related postponements or cancellations, over 80% occurred out of anxiety and fear of a COVID-19 infection [44]. We emphasize the consideration of the role of anxiety and concern due to COVID-19 in the context of delay in melanoma diagnosis [42].

Furthermore, we found that the increased Breslow thickness in females of older ages was mainly driven by nodular melanomas. Besides known associations of a worse outcome in nodular melanoma and increased thickness at diagnosis, a recent study suggested histopathological differences in melanoma subtypes with an independent risk factor for death in nodular melanoma [46]. Similar to our findings, an Italian study reported an increased proportion of nodular melanoma in the post-lockdown period [7], even though not further specified regarding gender aspects. In contrast to various previous observations [6,7,8], our data reveal no significant increase in tumor thickness among men as well as in the occurrence of ulceration as prognostic factors of melanomas in the post-lockdown period.

We observed that the proportion of AJCC stage IV melanomas in Switzerland increased significantly during the lockdown phase. Similar findings with proportionally more stage IIC-IV diagnosed melanomas were also observed in various previous studies from Greece [6] and the USA [47]. In this regard, Tejera-Vaquerizo et al. have built a lockdown-related upstaging model based on a linear rate of growth from stage T1 to T4 melanomas, estimating a 21% upstaging rate in the 1-month-delay-of-diagnosis group, 29% in the 2-month delay, and 45% in the 3-month-delay group [48]. Additionally, another model based on Australian melanoma registry data estimated an 8% upstaging shift after a 3-month delay and 32% after 6 months from T1 to T2 melanomas [49]. Regarding the impact of delays in diagnosis on different cancer survival outcomes due to pandemic restrictions, a modelling study from the UK estimated 3291–3621 additional deaths and a total of additional 59,204–63,229 years of life lost within 5 years in breast, colorectal, esophageal and lung cancer patients across different scenarios [11]. Contrary to our results, a higher proportion of early-stage melanomas was detected during the UK’s lockdown, where melanoma screenings were still provided even with ongoing restrictions, which highlights the necessity of providing skin checks [29]. It remains to be elucidated whether the higher-than-expected proportion of stage IV melanomas during lockdown in our study reflects a rapid upstaging to more advanced stages or rather missing early stages of thin melanomas. We postulate that the higher proportion of patients with more advanced melanoma diagnosed during the lockdown (Figure 4) may have been a consequence of patients with lower-stage melanoma postponing their hospital visit. Such postponement was observed in the higher rate of melanoma diagnoses directly after the removal of the major restrictions (Figure 1). Even though the EADV Melanoma Task Force encourages melanoma patients with stage 0-IIA to postpone their skin cancer screening for 2–3 months [19], we emphasize that all patients should be animated to perform self-examination. Multiple reasons may impact the delay of diagnosis during the pandemic, not only the withdrawal of skin cancer screenings, but also, e.g., reprioritization of services, teleconsultations and patient hesitancy.

### 4.3. Limitations

Multicenter studies are related to including only data from tertiary care hospitals because of a lack of available cancer registry data reflecting the total Swiss melanoma population. Regarding the COVID-19 incidence during the pandemic, there were also differences within Switzerland between German-, French-, and Italian-speaking regions, which were not taken into account in our analysis. Additionally, given the variety in the duration of lockdown and restrictions between countries, our findings need to be interpreted with caution.

## 5. Conclusions

The outbreak of COVID-19 imposes major challenges for outpatient care of melanoma patients. Our results provide clinically relevant insights about the potential effects of a lockdown-related diagnostic delay of melanomas. Lockdown restrictions resulted in gender- and age-dependent implications with older female patients being more likely to experience a deterioration of prognostic factors. The proportional increase in stage IV melanomas during lockdown supports the indispensability of an early detection of skin cancer also in times of crisis. Our findings suggest complex decision-making during the pandemic to be extremely cautious when dealing with cancer patients as well as to focus on gender- and age-specific implications in future pandemic management strategies.

## Figures and Tables

**Figure 1 cancers-14-02360-f001:**
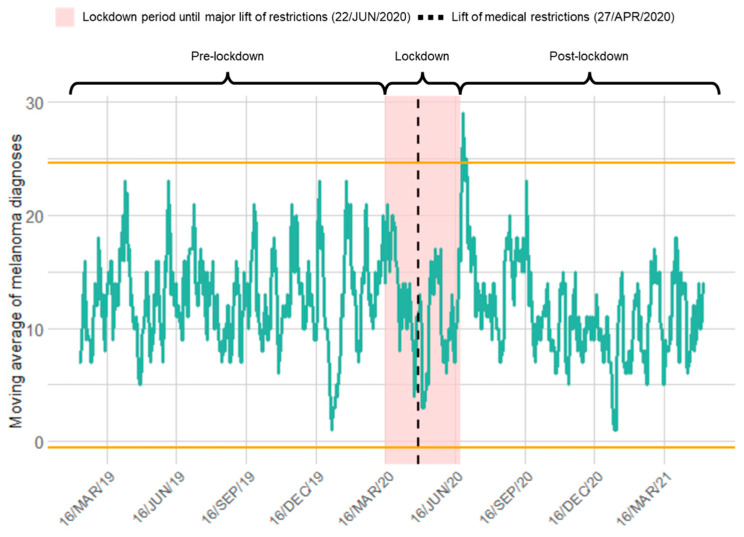
Time series of newly diagnosed melanomas relative to COVID-19 lockdown in Switzerland: A significantly higher rate of diagnosed melanomas was observed after the lift of major restrictions. The green line shows the rate, i.e., the moving 7-day average of diagnosed melanomas, considering a period of two years and three months centered on the official lockdown onset (16/MAR/20). The yellow lines represent three standard deviations above and below the mean of the considered interval. The shaded red area represents the lockdown period in Switzerland, defined here as the period between the official lockdown onset and the date for the major lift of restrictions (22/JUN/20). The black dashed line represents the date of the lift of medical restrictions (27/APR/20).

**Figure 2 cancers-14-02360-f002:**
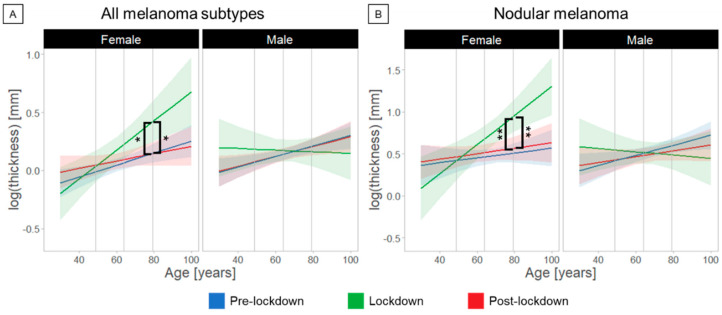
Age- and gender-dependent differences in melanoma diagnosis relative to the COVID-19 lockdown in Switzerland; post-hoc analysis showed increased tumor thickness in older women among all melanoma subtypes (**A**) and especially regarding the nodular melanoma subtype (**B**) during lockdown period. The logarithm of the melanoma thickness is plotted as function of age, gender, and lockdown periods. Asterisks indicate significant differences in the diagnosed melanoma thickness according to post-hoc analyses (* lockdown minus pre-lockdown in older female *p* = 0.03, * lockdown minus post-lockdown in older female *p* = 0.048; ** lockdown minus pre-lockdown in older female *p* = 0.0021, ** lockdown minus post-lockdown in older female *p* = 0.008). Note: Number of patients in A: female pre-lockdown: 271, lockdown: 62, post-lockdown: 164 and male pre-lockdown: 383, lockdown: 86, post-lockdown: 272; in B: female pre-lockdown: 43, lockdown: 14, post-lockdown: 32 and male pre-lockdown: 73, lockdown: 20, post-lockdown: 48.

**Figure 3 cancers-14-02360-f003:**
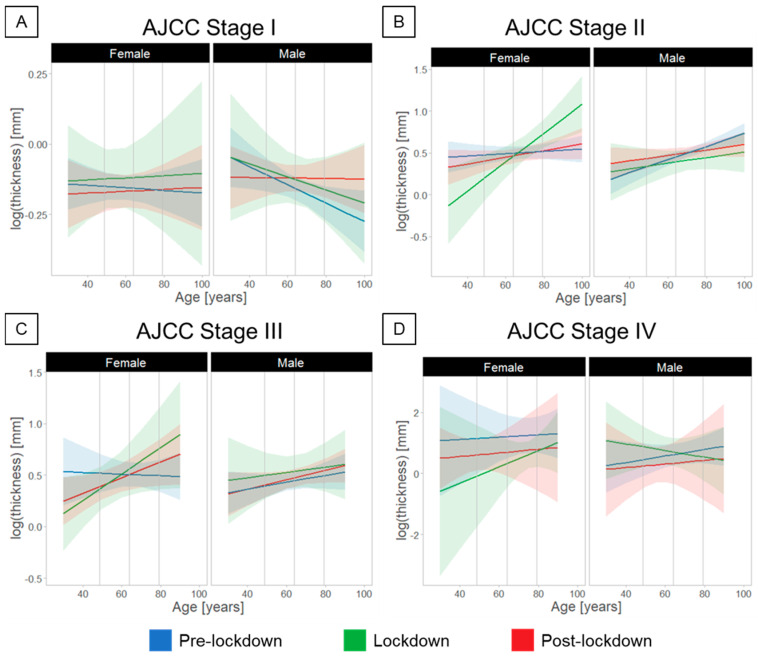
Age- and gender-dependent differences in the diagnosis of melanomas relative to the COVID-19 lockdown in Switzerland, considering stratification of melanomas in AJCC stages. (**A**) Stage I, (**B**) Stage II, (**C**) Stage III, (**D**) Stage IV. Effects of lockdown on the diagnosed melanoma thickness for older female patients were found only for stage II melanoma.

**Figure 4 cancers-14-02360-f004:**
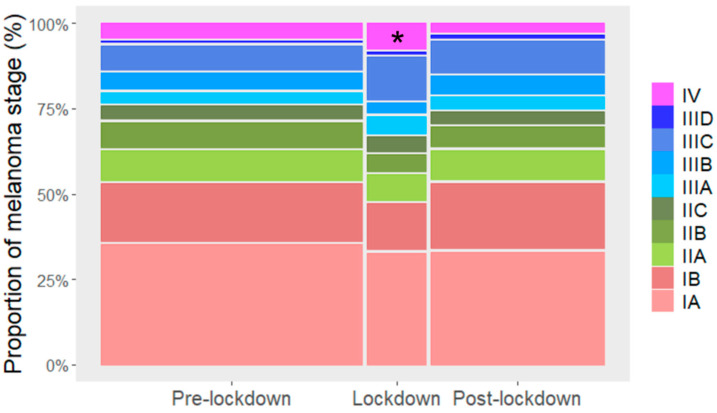
The proportion of the stages of diagnosed melanomas differed over time relative to the COVID-19 lockdown in Switzerland. The proportion of stage IV diagnosed melanomas was significantly higher during lockdown period (asterisk shows the cell with the largest residual). Chi-square test for stages grouped in I, II, III, and IV; *p* = 0.047).

**Figure 5 cancers-14-02360-f005:**
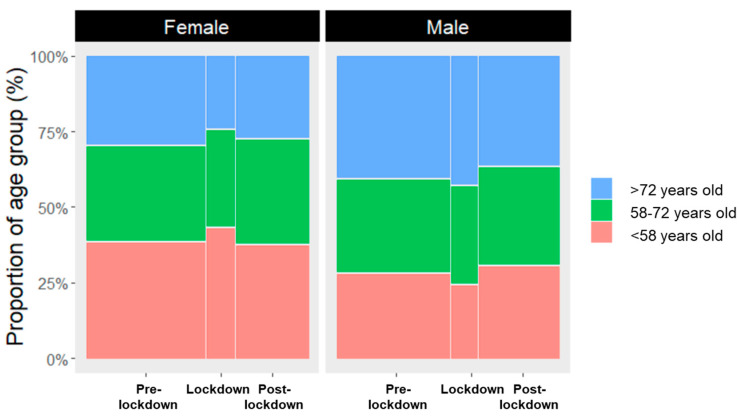
Age- and gender-dependent distribution of diagnosed melanomas relative to COVID-19 period in Switzerland. The proportion of diagnoses did not change before, during and after the COVID-lockdown period. Age groups are defined as young (<58 years), average (58–72 years) and old (>72 years).

**Table 1 cancers-14-02360-t001:** Characteristics of patients with invasive primary cutaneous melanoma.

Characteristics	n = 1240	
Mean age in years (SD)	64 (±15.4)	
Gender (n, %)	Female	497 (40.0)
	Male	741 (59.8)
	Not specified	2 (0.2)
Number of patients grouped by hospital in which diagnosis was received (n, %)	Aarau	152 (12.3)
	Basel	195 (15.7)
	Lucerne	130 (10.5)
	St. Gallen	184 (14.8)
	Zurich	579 (46.7)
Breslow thickness (median, (Q1–Q3)) [mm]	Total	1.2 (0.7–2.6)
	Women	1.1 (0.6–2.2)
	Men	1.4 (0.8–2.8)
	Pre-lockdown ^1^	1.2 (0.6–2.6)
	Women	1.1 (0.5–2.1)
	Men	1.4 (0.7–2.8)
	Lockdown ^1^	1.4 (0.7–3.0)
	Women	1.2 (0.7–3.0)
	Men	1.4 (0.7–3.0)
	Post-lockdown ^1^	1.2 (0.8–2.5)
	Women	1.2 (0.7–2.4)
	Men	1.3 (0.8–2.7)
Stage (8th edition AJCC; n, %)	Stage I	695 (57.0)
	IA	461 (37.2)
	IB	234 (18.9)
	Stage II	254 (20.5)
	IIA	115 (9.3)
	IIB	88 (7.1)
	IIC	51 (4.1)
	Stage III	233 (18.8)
	IIIA	46 (3.7)
	IIIB	63 (5.1)
	IIIC	114 (9.2)
	IIID	8 (0.7)
	III (not specified) *	2 (0.2)
	Stage IV	54 (4.4)
	Stage not specified	4 (0.3)
Ulceration (n, %)	Yes	240 (19.4)
	No	914 (73.7)
	Not specified	86 (6.9)
Melanoma subtype (n, %)	Superficial spreading	543 (43.8)
	Nodular	230 (18.5)
	Lentigo maligna	144 (11.6)
	Acral	43 (3.5)
	Others **	38 (3.1)
	Amelanotic	13 (1.0)
	Not specified	229 (18.5)

Note: ^1^ Pre-lockdown period = 01/FEB/19–15/MAR/20, lockdown period = 16/MAR/20–22/JUN/20, post-lockdown period = 23/JUN/20–30/APR/21. * Substage was not reported. ** Others = Spindle cell melanoma, blue naevus melanoma, ex naevus melanoma, desmoplastic melanoma, naevoid melanoma, mixed epithelioid spindle cell melanoma. N = sample size, SD = standard deviation, Q1 and Q3 = quartile 1 and quartile 3, resp.

**Table 2 cancers-14-02360-t002:** Significant pairwise differences for Breslow thickness in post-hoc analysis for significant interactions containing lockdown period as a factor.

**All Melanoma Subtypes** **(n = 1240)**						
** Interaction Period × Gender × Age—F(2,1182) = 3.15, *p* = 0.04 **						
**Period**	**Gender**	**Age**	**Difference ± SE [mm]**	**DoF**	**n**	***p*-Value**
Lockdown minus pre-lockdown	Female	Old	1.9 ± 1.3	207	333	0.03
Lockdown minus post-lockdown			1.9 ± 1.3	133	226	0.048
Pre-lockdown	Female minus male	Average	−1.2 ± 1.1	1183	654	0.045
Lockdown		Old	1.8 ± 1.3	1174	148	0.019
**Nodular Melanoma** **(n = 230)**						
** Interaction Period ** ** × Gender ** ** × Age ** ** — F(2,209) ** ** = 4.49, *p* ** ** = 0.012 **						
**Period**	**Gender**	**Age**	**Difference ± SE [mm]**	**DoF**	**n**	***p*-Value**
Lockdown minus pre-lockdown	Female	Old	2.9 ± 1.3	39	57	0.0021
Lockdown minus post-lockdown			2.6 ± 1.4	41	46	0.008
Lockdown	Female minus male	Average	1.7 ± 1.3	196	34	0.03
		Old	3.3 ± 1.4	133	34	0.0013
**AJCC Stage II** **(n = 254)**						
** Interaction Period ** ** × Gender ** ** × Age ** ** — F(2,173) ** ** = 3.39, *p* ** ** = 0.04 **						
**Period**	**Gender**	**Age**	**Difference ± SE [mm]**	**DoF**	**n**	***p*-** **Value**
Lockdown	Female minus male	Old	2.2 ± 1.4	160	27	0.016

Note: Black cells represent inferential statistics from significant three-way interactions Period × Gender × Age in the Breslow thickness of diagnosed melanoma for all melanoma subtypes (top), as well as only for nodular melanomas (middle) and for melanomas classified as stage II (bottom), according to the AJCC. The white cells represent significant pairwise differences for Breslow thickness in post-hoc analyses following the significant interactions reported in the black cells. SE = standard error of the model; DoF = degrees of freedom; n = sample size.

## Data Availability

Fully anonymized data can be requested from the corresponding author.

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
