# Peer review of "Effects of COVID-19 Lockdown on Melanoma Diagnosis in Switzerland: Increased Tumor Thickness in Elderly Females and Shift towards Stage IV Melanoma during Lockdown"

_cancers, 2022, doi:10.3390/cancers14102360_

Round 1
Reviewer 1 Report
Methods:
Line 93 and line 126: Please clarify the exact time of lockdown, 42 days are mentioned first but then the period 16/MAR/20 - 22/JUN/20 is used? In lines 181/182 in results section, then suddenly 3 months before during and after lock down are reported? This is very confusing to report all these different time periods, and the reviewer is left feeling unsure which one is used when? Why not just focus on the about similar number of patients and diagnoses during these three 3-months periods? This would be much cleaner than the current varied time periods dotted throughout the manuscript? For example, which time eriods were used in Table 1?.
Line 100: Given that the time periods investigated vary widely the “number of melanoma diagnoses” cannot be used, rates must be calculated – which is probably what was done as the 7 day moving average rate is mentioned later on in lines 142/143.
Line 168: the expression “dependence of the number of classified stages of diagnosed melanoma” is not very clear please consider rephrasing?
Line 171: what was the rationale for the age cut-off chosen?
Line 172: the sentence “The time of diagnosis contained the levels pre-lockdown, lockdown, and 172 -lockdown, as defined previously” is not clear how can a time contain a period? Please rephrase.
Results:
How complete do the 1,240 melanoma patients represent the eligible population? Have any melanoma patients been missed? How was quality assurance doe to be certain none were overlooked?
Line 223: the expression “reached levels above normality” is unclear, please revise the working I assume that you mean a greater number of diagnosis? But you may also mean a larger thickness or stage? In the next line you then refer to ‘rate’. I suggest that throughout the manuscript the rate/10,000 or similar is reported, so that the same metrics is used.
Line 226: the terminology “in the time course of melanoma diagnosis” is unclear and needs to be clarified.
Figure 1: while this shows a drop and spike around the time of the lock down, statistical analysis is required to assess whether tis is out of range, probably using methods of process control. See review of such methods: https://www.ncbi.nlm.nih.gov/pmc/articles/PMC2464970/
Line 266: what period was used in these analyses?
Table 2: The table is quite complex and not easy to understand what was done. It needs to contain the n for each of the cells to be able to understand the comparisons at least.
Figure 3: Doe it make sense to have more stage 4 cancers during lockdown? Would you not have expected this to happen after lockdown if diagnosis was delayed? It looks as if there were fewer stage 4 diagnoses after lockdown? Can it be clarified if this is numbers of rates presented in Figure 3?
Reviewer 2 Report
The paper ”Effects of COVID-19 lockdown on melanoma diagnosis in Switzerland: Increased tumor thickness in elderly females and shift towards stage IV melanoma during lockdown” by Kostner, Cerminara, Pamplona, and colleagues describes a well-defined, retrospective, exploratory cohort study, stratified by three time periods. The presentation is clear, the argumentation easy to follow, and the conclusions well-balanced in light of the results given. The graphical illustrations are exemplary. Please find a few comments and suggestions for further improvement below.
Line 32, 108, 338. Please add as follows: “In this retrospective, exploratory cohort study, …”
Line 34-35. Please add the distribution of the included 1,240 patients across the three strata in the abstract; e.g. “…the pre-lockdown (01/FEB/19-15/MAR/20; n=XX) with the lock-34 down (16/MAR/20-22/JUN/20; n=YY) and post-lockdown period (23/JUN/20–30/APR/21; n=ZZ)”.
Graphical abstract, Figure “Age- and gender-dependent…”. Please add the respective number of patients for both males and females in both parts of the figures that is (A) and (B).
Line 159-160. Please specify the age intervals more accurately, for instance as follows. “Young”: smaller than mean age minus one standard deviation; “average”: larger than mean age minus one standard deviation and smaller than mean age plus one standard deviation (or: within one standard deviation around mean age); “old”: larger than mean age plus one standard deviation.
Line 171. Replace “quantiles” by “strata” (or, “subgroups”)
Line 177. In light of the exclusion criteria laid out in l.111-112, how many patients were excluded? Please add this number to underline that probably only a minor proportion was disregarded, thanks. Usually, a flowchart depicts the patient flow of and within the study; however, this is probably dispensable for this retrospective study.
Line 182-183. Please add the number of patients for each time interval and respective percentages to enhance comparability across time frames; “…before lockdown was 143 out of XXX (xx%), during the three-months lockdown 149 out of YYY (yy%), and 153 out of ZZZ (zz%) three months after lockdown.”
Figure 2. In both (A) and (B), there is a superfluous parenthesis and a superfluous asterisk, please delete.
Lines 295, 332. Please put concrete p-values into the text, not just <0.05 or <0.01; p-values should be interpreted continuously, see, for instance, Sterne JA, Davey Smith G. Sifting the evidence-what's wrong with significance tests? BMJ. 2001 Jan 27;322(7280):226-31. doi: 10.1136/bmj.322.7280.226.
Line 311, 327. I propose to move both supplemental figures into the main text, which, then, will become Fig. 3 and Fig. 5, respectively (current Fig. 3 [referenced in l.326] becomes Fig. 4). Both figures are supportive illustrations for the main text.
Line 318. Replace “(all ps>0.05).” simply by “(p>0.05).”
Line 368. Delete “made” (to make the sentence read “…restrictions were imposed…”).
Line 416. Delete “(2020)”
Line 417. Delete “[7]” (as it also appears in line 418 at the end of the sentence).
Line 501. Replace “in” by “for”
References. Please revisit the Cancers’ Information for Authors and make sure that ALL references are formatted accordingly and appropriately. Regarding, for instance, your reference #1: I reckon that all authors (or at least more than one) should be listed when using “et al.”; journal abbreviations employ full stops (Sci. Prg.); year of publication should be in bold print, the journal’s volume in italic print; no “p.” is used in front of the page numbers. See, for instance, also your reference #39 (“issue: 1,): p. page(s)”).
Round 2
Reviewer 1 Report
The authors have addressed the reviewer comments well.